# Constraints on Stress Tensor from Microseismicity at Decatur

Tian Guo<sup>1</sup> <sup>2</sup>, Dmitry Alexandrov<sup>3</sup>, Leo Eisner<sup>3</sup>, Zuzana Jechumtalova<sup>3</sup>, Sherilyn Coretta Williams-Stroud<sup>4</sup>, Umair Bin Waheed<sup>5</sup>, Víctor Vilarrasa<sup>1</sup>

- <sup>1</sup>Global Change Research Group (GCRG), Mediterranean Institute for Advanced Studies, Spanish National Research Council (IMEDEA-CSIC-UIB), Esporles, 07190, Spain.
- <sup>2</sup>Department of Civil and Environmental Engineering (DECA), Universitat Politècnica de Catalunya (UPC Barcelona Tech), Barcelona, 08034, Spain
- <sup>3</sup>Seismik s.r.o., Prague, 18200 Praha 8, Czech Republic
- <sup>4</sup>Department of Geology & Geophysics, Texas A&M University, Texas 77843-3115, United States
- 10 5Department of Geosciences, King Fahd University of Petroleum & Minerals, Dhahran 31261, Kingdom of Saudi Arabia

Correspondence to: Tian Guo (tian.guo@csic.es)

Abstract. Induced microseismicity has been detected in the Decatur CO<sub>2</sub> sequestration area, providing critical constraints on the stress state at the reservoir. We invert the full stress tensor with two subsets of source mechanisms from the induced microseismic events. To achieve this, we incorporate additional information on the vertical stress gradient and instantaneous shut-in pressure (ISIP) measured in the area. Additionally, our results demonstrate that constraining the intermediate stress tensor to a vertical orientation is essential to achieve a consistent stress inversion. The inverted stress is then used to estimate the minimum activation pressure required to trigger seismicity on fault planes identified by the source mechanisms. The comparison of the minimum activation pressure with injection pressure indicates one of three possibilities: the ISIP pressures are significantly lower than predicted (approximately 28-29 MPa), the maximum horizontal principal stress is extremely high (exceeding 120 MPa), or the coefficient of friction is significantly lower than 0.6 on a large number of activated faults. Our analysis also shows that poorly constrained source mechanisms do not lead to reasonable stress constraint estimates, even when considering alternative input parameters such as ISIP and vertical stress. We conclude that induced microseismicity can effectively be used to estimate the stress field when source mechanisms are also well constrained. For future CO<sub>2</sub> sequestration projects, measuring and constraining ISIP pressure and maximum horizontal stress in the reservoir will ensure that more accurate estimates of stress state from moment tensor inversions can be obtained for improved prediction of the long-term reservoir response to injection.

#### 1 Introduction

The Illinois Basin Decatur Project (IBDP) is a large-scale carbon capture and storage (CCS) project designed to sequester supercritical CO<sub>2</sub> derived from biofuel sources (Goertz-Allmann et al., 2017). Between November 2011 and November 2014, 999,215 tonnes of supercritical CO<sub>2</sub> were successfully injected, with an average injection rate of about 1,000 tonnes per day (Greenberg et al., 2017). Microseismicity is a common consequence of fluid injection into reservoirs. Given the relatively large scale and high injection rate at the IBDP site, the potential for injection-induced microseismicity was a concern and was consequently integrated into the project's design (Bauer et al., 2016). Passive seismic monitoring started 564 days before the

injection, during which over 68,000 triggered events were recorded, most of which were related to drilling or other well operations (Smith and Jaques, 2016). During and after the injection period, around 20,000 microseismic events were detected by the downhole array, with magnitudes ranging between -2.1 and 1.2 (Langet et al., 2020; Williams-Stroud et al., 2020). Locatable microseismic events generally started two months after injection started, and tends to cluster (Bauer et al., 2016; Kaven et al., 2015).

Understanding the stress regime within the Illinois Basin is crucial for assessing the rock's response to injected fluids, which can increase pore pressure and induce seismicity (Lahann et al., 2017). The In Salah CCS Project (Goertz-Allmann et al., 2014) increased the injection pressure by 32% to 80% over the original formation pressure. At the IBDP site, the maximum increased pressure was measured at 1.14 MPa in the monitoring well VW1, which was about 5.2 % above the initial pore pressure (around 20 MPa), representing only 65% of the fracture pressure for the Mt. Simon injection zone, implying that the injection does not fracture the formation and should not change the background stress orientation (Bauer et al., 2016).

## 45 1.1 Study site - IBDP

The IBDP is an integrated bioenergy CCS project permanently storing CO<sub>2</sub>, which is a byproduct of ethanol production at the Archer Daniles Midland (ADM) facility (Kaven et al., 2015). The IBDP site is located in the north-central region of the Illinois Basin in east-central Illinois, in the central United States (Bauer et al., 2016). This project was led by the Illinois State Geological Survey (ISGS).

The primary injection target during this stage of injection was the lower part of the Cambrian Mt. Simon sandstone, a regionally extensive formation within the Illinois Basin that is of variable thickness (locally approximately 460-m thick), the porosity averages 22% and average permeability is 200 mD but can be as high as 1000 mD (Leetaru et al., 2014). This widespread formation beneath much of the Midwestern United States, which is also used locally for geological storage of natural gas (Bauer et al., 2019). The Mt. Simon is divided into three major sections - Lower, Middle, and Upper- and is overlain by the Cambrian Eau Claire, which serves as the primary caprock, with two additional sealing formations in the sedimentary sequence above (Leetaru and Freiburg, 2014). The Cambrian Eau Claire shale formation has a thickness of about 152 m and low permeability; the upper half of the Eau Claire grades from limestone to clayey limestone to the very shaley lower half of the formation that serves as an effective seal (Leetaru and Freiburg, 2014; Williams-Stroud et al., 2020). The Argenta formation underlies the Mt. Simon formation and consists of compact sandstones and pebbly conglomerate with clay cement and angular clasts of bedrock, with a low average porosity of 9% and low average permeability about 2 mD (Leetaru et al., 2014).

During the injection phase of IBDP, 999,215 tonnes of supercritical CO<sub>2</sub> were successfully injected into the lower Mt. Simon sandstone (Greenberg et al., 2017). The depth of the injection interval ranged from 2129.6 to 2149.4 meters with an average downhole injection pressure of 23.0 MPa (Will et al., 2016). The average injection pressure provides an important constraint on the pore pressure, as the pore pressure that activated induced seismicity is not expected to significantly exceed this value.

We know that the injection triggered these events and the events are far from the injection well. Because of this considerable distance, thermal effects can be neglected as a triggering mechanism. Hence the most likely the triggering mechanism is increased pore pressure on pre-existing faults or fractures.

70

Figure 1(a) shows the major project elements of the IBDP site. As Figure 1(b) shows, the injection well CCS1 was drilled to a depth of 2206 m (1.92–1.94 km below sea level), terminating in the crystalline basement. Three multicomponent geophone arrays were installed in this well. The verification well (VW1) was designed for deep reservoir monitoring, located approximately 307 m north of CCS1 and drilled to a depth of 2217 m. The geophysical monitoring well 1 (GM1) located 60 m west of CCS1 and reaches the depth of 1067 m, it was instrumented with 31-level three-component geophone array (2 close to the surface, with 29 of these geophones at depths between 418 m and 844 m) installed for Vertical Seismic Profile (VSP) and also for microseismic monitoring (Goertz-Allmann et al., 2017). 18 surface sensors were installed by U.S. Geological Survey (USGS) and ISGS 18 months after the start of injection.

Figure 1: (a) The IBDP site showing the location of major project elements. The yellow outline for the demonstration site is approximately 800 m (2600 ft) on a side. The blue rectangle in the southeast corner of the image is the compressor (Modified from

Finley, (2014), © Society of Chemical Industry and John Wiley & Sons, Ltd. Reproduced with permission. Aerial imagery originally acquired by the Illinois Department of Transportation (IDOT), Aerial Surveys Division, as part of the Illinois Basin – Decatur Project (IBDP)). (b) 3D schematic of the monitoring and injection system at the IBDP site. The layout shows geophone arrays, well trajectories (VW1, GM1, CCS1), and major geological units. Adapted from Illinois State Geological Survey (ISGS) et al. (2021).

#### 1.2 Stress state at the Decatur site




The overall stress regime for the Illinois Basin is primarily characterized by strike-slip faulting, with some reverse faulting, based on earthquake focal mechanisms, borehole stress indicators and in-situ stress measurements (Zoback and Zoback, 1989). Using the Anderson (1905) classification, the corresponding stress configuration can be defined as strike-slip, where  $S_{Hmax} > S_V > S_{hmin}$ .

The vertical stress ( $S_v$ ) is typically determined by integrating density logs from the surface to the target area; however, there are challenges including creating a reliable density-depth profile due to limited shallow density data and variable lithology at the sediment surface in Illinois (Lahann et al., 2017). Bauer et al. (2016) estimated the vertical stress gradient at the IBDP site to be approximately 23.75 MPa/km using the integrated density method. Lahann et al. (2017) generated local  $S_v$  profiles with a boundary at 2134 m, estimating gradients of 27.1 MPa/km for the deeper central basin and 25.1 MPa/km for shallower depths. Langet et al. (2020) summarized the values from published papers and Illinois State Geological Survey (ISGS) reports, setting a mean value of 25 MPa/km with a standard deviation of 2 MPa/km for the vertical stress and an uncertainty of approximately 8%.

The minimum horizontal stress ( $S_{hmin}$ ) was estimated by in-situ formation strength test data from oil and gas wells in the Illinois Basin (Lahann et al., 2017). According to Bauer et al. (2016), the  $S_{hmin}$  was evaluated from the borehole testing results, ranging from 31.6 to 34.2 MPa at a depth of 2.14 km in the Mt. Simon formation. This range is lower than the calculations from Lahann et al. (2017), who assessed stress gradients from multiple data sources, including fracturing records from wells in the project area, suggesting values between 24.0 and 27.3 MPa/km. Langet et al. (2020) reported a mean value of 22.8 MPa/km with a standard deviation of 4 MPa/km for  $S_{hmin}$  and an uncertainty of about 17.5%. We use these gradients to estimate stress magnitudes, assuming a linear relationship for the entire depth range, where stress is computed. Specifically,  $S_{hmin}$  and  $S_v$  were calculated by multiplying the average depth of each dataset by the corresponding gradient value.

The maximum horizontal stress ( $S_{Hmax}$ ) was estimated using the critically stressed fault model (Lahann et al.,2017), with values calculated from hydrofracture data assuming a fluid pressure of 10.2 MPa/km and varying frictional coefficients from 0.6 to 1.0 (Lahann et al.,2017). This approach produces a wide range of  $S_{Hmax}$  gradient values (from 40 MPa/km to 82.6 MPa/km), with significant high relative uncertainty of 35.2% (Lahann et al.,2017). Langet et al. (2020) summarized a mean  $S_{Hmax}$  value of 61.3 MPa/km with a standard deviation of 15 MPa/km and uncertainty of about 24.5%. The  $S_{Hmax}$  azimuth, between 060° and 075–080°, aligns with the stress directions of the eastern North American plate (Bauer et al.,2016, Lahann et al., 2017).

Overall, the  $S_{Hmax}$  carries the highest uncertainty of about 35.2%, reflecting the general challenge of measuring  $S_{Hmax}$ .  $S_{hmin}$  is better constrained with approximately 17.5% uncertainty, due to the availability of multiple data sources that provide direct measurements of reservoir strength (Bauer et al. 2016). The best constraint is on the vertical stress  $S_v$  estimations supported by more extensive data from experiments and reports.

In this study, the stress gradient values are applied to the average depth of different source mechanism datasets at the IBDP site to estimate vertical stress and minimum horizontal stress. These stress estimates are then used to constrain the full stress tensor in the volume where microseismic events occurred, together with the source mechanisms of the induced events. The range of maximum horizontal stress estimated by the stress gradient values is compared to the inversion results to evaluate the accuracy of the stress inversion. The full stress tensor is then used to constrain the minimum activation pressure required to trigger seismicity on fault planes identified by the source mechanisms.

#### 2 Microseismic data -Stress constraint



The anticipated induced seismicity was observed at IBDP, while not in locations coincident with the fault interpreted from activated seismic data (Williams-Stroud et al., 2020). We interpret the occurrence of induced seismicity as evidence that the stress state or pore pressure in the reservoir was sufficiently perturbed to trigger slip. Additionally, we analyze the source mechanisms of the events to further constrain the stress state in the Decatur Basin. As shown in Figure 2, we use two sets of source mechanisms of microseismic events. The locations of induced microseismic events from these datasets are approximately two kilometers north of CCS1, and they occurred during the injection period at CCS1 well. The source mechanisms of only 339 events were determined, with those of other induced events either not known or poorly constrained due to low signal to noise ratio (SNR). The majority of the selected events are far enough from the injection to exclude the possibility that the low-pressure injection reoriented background stress.

Figure 2: Map of source mechanisms of selected induced events in Decatur reservoir. The black triangles represent surface stations (USGS/ISGS), the black star represents injection well CCS1, black circles present surface projection of downhole monitoring array receivers. The color coding of the source mechanism represents the depth from the mean sea level of selected seismic events.

Dataset 1 includes 23 selected microseismic events from the northern cluster that were published by Langet et al. (2020) (Fig. 3(a)). This cluster is one of the major clusters characterized by Goertz-Allmann et al. (2017). The locations of events were determined in a 1D velocity model, incorporating seismic data from the downhole geophone array and additional monitoring receivers from USGS and ISGS networks to improve the data accuracy. The combined use of surface and downhole sensors significantly enhance the station coverage over the focal sphere (Langet et al., 2020). The source mechanisms of the events were inverted using P-wave amplitude and polarities from manually picked P-waves on vertical components of the downhole arrays. The inversion was constrained to a shear mechanism using grid search over strike, dip, and rake angles (Langet et al., 2020).




Dataset 2 includes 26 most suitable events located throughout the reservoir is obtained from electronic supplement of Williams-Stroud et al. (2020) (Fig. 3 (b)). Events were located using data from surface stations of USGS and ISGS, with a consistent number of 12 stations utilized per event. The source mechanisms were inverted from both P- and S-wave amplitudes using a 1D velocity model and attenuation model (Staněk and Eisner, 2017). The number of P-wave picks for these inversions ranged from 6 to 16, and S-wave picks also ranged from 6 to 16.

Figure 3: Maps of source mechanisms of selected induced seismic events of Dataset 1 and Dataset 2. Left plot (a) shows Dataset 1 (Langet et al. ,2020), the right plot (b) shows Dataset 2 (Staněk et al. ,2019).

Both datasets represent similar events (the best SNR) and 11 events of Dataset 1 are also part of Dataset 2, but the source mechanisms, depths, and epicentral positions of these events significantly differ. Note that Dataset 1 consists mostly of strike-slip events, while Dataset 2 is mostly strike-slip, but also dip-slip and reverse events. The depth differences of the hypocenters of the corresponding events differ up to 750 m but epicentral differences of corresponding events are smaller than 500 m. The differences result from variations in velocity and attenuation models, methodologies of source mechanism inversion, and the monitoring arrays used. Specifically, Dataset 1 utilized both surface stations and downhole geophones data (Langet et al., 2020), providing a more enhanced array with 3D azimuthal coverage, whereas Dataset 2 had different station configurations, primarily relying on surface stations (Williams-Stroud et al., 2020). Consequently, the focal coverage for the events in Dataset 2 mainly constrained the horizontal distribution of ray path, which may lead to less constrained vertical control and ambiguities in the focal mechanism solution. We cannot assume the depths of Dataset 1 are better constrained because they used the downhole monitoring array, which is far and does not span the depth of monitored events. The differences in source mechanisms result from uncertainties in the seismic data and velocity model, and represent inherent uncertainty in the inverted source mechanisms. We use this empirically estimated uncertainty of these mechanisms in the inversion process and set this uncertainty 5° and 10° in dip, strike, and rake angles.

Table 1 General information of the Dataset 1 and 2




| Dataset | Number of events | Start Date  | End Date    | Ma   | agnitude (M <sub>w</sub> ) | Depth (m.b.s.l) |
|---------|------------------|-------------|-------------|------|----------------------------|-----------------|
| 1       | 23               | 2014-Jul-03 | 2015-Feb-07 | Max  | 0.76                       | 1955            |
|         |                  |             |             | Mean | 0.29                       | 1950            |

|   |    |             |             | Min  | 0.03  | 1910 |  |
|---|----|-------------|-------------|------|-------|------|--|
| 2 | 26 | 2013-Dec-26 | 2014-Nov-23 | Max  | 0.86  | 2889 |  |
|   |    |             |             | Mean | 0.20  | 2545 |  |
|   |    |             |             | Min  | -1.04 | 2365 |  |

# 165 3 Methodology


## 3.1 General description methodology

To determine the pore pressure that caused induced seismicity, we need to know the shear and normal stress on the fault plane, the coefficient of friction and the cohesion on the faults of microseismic events. Calculating the shear and normal stresses on a fault plane requires knowledge of the full stress tensor. Given the large uncertainty on the stress tensor magnitudes and orientation in Decatur reservoir, we determine stress tensor consistent with observed source mechanisms in each dataset. To invert the full stress tensor, we need additional borehole data – in our case, ISIP and vertical stress. With ISIP and vertical stress and their uncertainties, the minimal activation pressure is estimated for each source mechanism using the inverted stress tensor and the Mohr-Coulomb fault failure criterion when assuming zero cohesion and a constant friction coefficient of 0.6.

Figure 4: General description of minimum stress determination.

#### 175 3.2 Stress Inversion

We use the source mechanisms with estimates of ISIP and vertical stress to carry out the joint full stress inversion algorithm. The stress orientation is well constrained by the earthquake mechanisms and the magnitudes are constrained by additional borehole data and the shape ratio resulting from source mechanisms.

The algorithm is based on the modified methodology of Gephart & Forsyth (1984) and Angelier *et al.* (1982) as we assume that the seismic slip occurs in the direction of shear stress acting on pre-existing faults or fractures (Bott, 1959; Wallace, 1951)(Wallace-Bott hypothesis). The stress inversion procedure is based on two assumptions:

- 1. There exists one stress field solution that describes all available data within uncertainty.
- 2. Slip vector, acting on the fault plane of each focal mechanism, is parallel to the tangential stress acting on that plane.

## 3.2.1 Stress inversion algorithm

The stress tensor  $(\sigma_{ij})$  is a symmetric second-order tensor with six independent components due to the equilibrium condition, it can be defined by the stress tensor matrix  $\hat{\sigma}$ . Its principal stresses  $(\sigma_1, \sigma_2, \sigma_3)$  correspond to the eigenvalues of the matrix, with eigenvectors defining the stress state in a coordinate system where the tensor is diagonal. The full stress tensor can be decomposed into a hydrostatic and a deviatoric component. When focal mechanisms are studied, it is common to work only with deviatoric stress tensor by isolating the hydrostatic part of the stress tensor matrix Eq. (1):

$$\hat{\sigma} = \frac{1}{3} I \operatorname{tr}(\hat{\sigma}) + \left(\hat{\sigma} - \frac{1}{3} I \operatorname{tr}(\hat{\sigma})\right), \tag{1}$$

where I is the identity matrix,  $tr(\hat{\sigma}) = \sigma_{11} + \sigma_{22} + \sigma_{33}$  is the trace of the matrix  $\hat{\sigma}$ , and the term  $\left(\hat{\sigma} - \frac{1}{3}I tr(\hat{\sigma})\right)$  represents the deviatoric stress.

In order to fulfil the Wallace-Bott hypothesis for a focal mechanism, the stress vector  $(T = \hat{\sigma} n)$  acting on a fault plane should have components only along the normal to the fault plane n and along the slip vector s, the stress tensor must satisfy Eq. (2):

$$\|\hat{\sigma}\mathbf{n}\|^2 = (\hat{\sigma}\mathbf{n}, \mathbf{n})^2 + (\hat{\sigma}\mathbf{n}, \mathbf{s})^2, f = (\hat{\sigma}\mathbf{n}, \mathbf{s})^2 - \sqrt{\|\hat{\sigma}\mathbf{n}\|^2 - (\hat{\sigma}\mathbf{n}, \mathbf{n})^2} = 0.$$
 (2)

For the whole set of N focal mechanisms, the vector of constraints can be defined as  $f = (f^{(1)}, f^{(2)}, ..., f^{(N)})^T = 0$ , Angelier et al. (1982) minimize the quadratic form  $S = (\pi - \pi_0)^T \cdot C_0^{-1} \cdot (\pi - \pi_0)$  under the constraint  $f(\pi) = 0$ , where  $\pi$  is a vector that constrains unknown deviatoric stress parameters as well as focal mechanisms parameters,  $\pi_0$  contains focal mechanism data and a priori estimate of the stress tensor,  $C_0$  is the covariance matrix of  $\pi_0$ . The constraint function  $f(\pi)$  has Fréchet's derivative matrix  $(F_n)_{jk} = \left(\frac{\partial f_j}{\partial \pi_k}\right)\Big|_{\pi=\pi_n}$  at point  $\pi_n$  for all unknowns and data. The solution can be then found iteratively using non-linear least-squares inversion (Tarantola and Valette, 1982) according to Eq. (3):

$$\mathbf{\pi}_{n+1} = \mathbf{\pi}_0 + C_0 \cdot F_n^T \cdot (F_n \cdot C_0 \cdot F_n^T)^{-1} \cdot [F_n \cdot (\mathbf{\pi}_n - \mathbf{\pi}_0) - f(\mathbf{\pi}_n)], \tag{3}$$

The inversion process modifies not only a priori values of the stress tensor, but also the focal mechanisms (dip, strike, and rake) within the uncertainty of the inverted mechanisms.

#### 205 3.2.2 Inversion with additional stress constraints


With vertical stress based on density logs and minimum principal stress estimated from leak-off pressure (LOP) or ISIP, it is possible to find the full local stress tensor after inverting focal mechanism (unless minimum principal stress direction is

vertical). This can be done in two steps without modifying the discussed inversion methods by first estimating the deviatoric stress, and then computing the remaining stress magnitudes  $\sigma_1$  and  $\sigma_2$  using the shape factor  $R = \frac{\sigma_2 - \sigma_1}{\sigma_3 - \sigma_1}$  and Eq. (4):

$$\sigma_{33} = \sigma_1 P_{31}^2 + \sigma_2 P_{33}^2 + \sigma_3 P_{33}^2, \tag{4}$$

where  $P_{ii}$  is a rotation matrix based on estimated Euler angles, that defines stress tensor in the geographical frame Eq. (5):

$$\hat{\sigma} = P \begin{pmatrix} \sigma_1 & 0 & 0 \\ 0 & \sigma_2 & 0 \\ 0 & 0 & \sigma_3 \end{pmatrix} P^T.$$
 (5)

Unless the minimum principal stress is vertical, in which case  $P_{31} = P_{32} = 0$  and  $P_{33} = 1$ , the solution for principal stress magnitudes is given by Eq. (6):

$$\sigma_{1} = \frac{\sigma_{33} - \sigma_{3}(P_{32}^{2}R + P_{33}^{2})}{P_{31}^{2} + P_{32}^{2}(1 - R)},$$

$$\sigma_{2} = \frac{(1 - R)\sigma_{33} + \sigma_{3}(RP_{31}^{2} - (1 - R)P_{33}^{2})}{P_{31}^{2} + P_{32}^{2}(1 - R)},$$
(6)

$$\sigma_3 = P_{ISIP}$$

#### 3.3 Full stress inversion


In order to find the stress orientation and shape factor R using a set of focal mechanism, the misfit function is built as Eq. (7):

$$S(R,\xi_1,\xi_2,\xi_3) = \sum_j \frac{\left|\chi_{min}^{(j)}\right|}{\delta\alpha_j},\tag{7}$$

where  $\xi_1, \xi_2$  and  $\xi_3$  are the Euler angles defining stress tensor orientation in the geological frame (North, East and the center of the earth),  $\chi_{min}^{(j)}$  is the minimal rotation angle of the *j*-th mechanism and  $\delta \alpha_j$  represents uncertainty of the *j*-th mechanism. The normalization of rotation angle with uncertainty ensures that poorly defined focal mechanisms contribute less to the misfit function. The angle  $\chi_{min}^{(j)}$  is minimal along six angles of rotation: three angles for each nodal plane and for each focal mechanism we keep the plane that  $\chi_{min}^{(j)}$  corresponds to. The fault plane is identified for each mechanism once the sum is minimized. Full stress inversion is performed on the basis of the method of Gephart & Forsyth (1984) and non-linear squares inversion of Angelier et al. (1982). The former method yields an approximate solution and identifies the fault planes, while the latter refines the approximate solution and utilizes the information about the fault planes.

The first step is to modify the misfit function of the Gephart & Forsyth (1984) approach as Eq. (8):

$$S = \sum_{i}^{N_{\text{mech}}} \frac{\left| \chi_{\min}^{(j)} \right|}{\delta \alpha_{j}} + \frac{\left| \sigma_{33} - \sigma_{v} \right|}{\delta \sigma_{v}} + \frac{\left| \sigma_{3} - P_{ISIP} \right|}{\delta P_{ISIP}}, \tag{8}$$

where the first term represents source mechanism inversion - the sum of the minimal rotation angles that make slip parallel to the tangential stress for each of  $N_{\rm mech}$  focal mechanisms normalized by the mechanism uncertainty  $\delta\alpha_j$ ,  $\sigma_v$  and  $P_{ISIP}$  are measured weight of overburden and ISIP value,  $\delta\sigma_v$  and  $\delta P_{ISIP}$  are corresponding uncertainties,  $\sigma_{33}$  and  $\sigma_3$  are values from the candidate solution.

- 235 Minimization of the misfit function is carried out using the stochastic optimization technique proposed by Yang & Deb (2009), the so-called Cuckoo Search, which is a biologically inspired optimization algorithm and simulates breeding behavior of certain species of cuckoos and performs random walks between candidate solutions with the step length draw from the Lévy distribution. This robust algorithm has provided optimization on various standard optimization test problems.
- After this inversion step, inconsistent mechanisms are dropped, i.e., the corresponding minimal rotation angle  $\chi_{\min}^{(j)}$  is larger than mechanism uncertainty  $\delta \alpha_j$ . Then nonlinear least square inversion according to Eq. (3) is proceeded. The result of the extended Gephart and Forsyth (1984) algorithm is the initial solution  $x_0$ , in which the vector of constraints f is defined as Eq. (9) to include additional stress constraints:

$$f = \begin{pmatrix} \left(\hat{\sigma}\boldsymbol{n}^{(j)}, \boldsymbol{s}^{(j)}\right) - \sqrt{\|\hat{\sigma}\boldsymbol{n}^{(j)}\|^2 - (\hat{\sigma}\boldsymbol{n}^{(j)}, \boldsymbol{n}^{(j)})^2} \\ \sigma_{33} - \sigma_v \\ \sigma_3 - P_{ISIP} \\ 1 \le j \le N_{\text{mech}} \end{pmatrix}$$
(9)

The above defined stress inversion allows to include uncertainty of both inverted source mechanisms as well as additional borehole measurements.

#### 3.4 Pore pressure calculation


Finally, after obtaining the full stress, we can utilize focal mechanisms to determine pore pressure that caused the microseismic event to occur. For this purpose, we utilize the Mohr-Coulomb fault failure criterion, which is given as Eq. (10):

$$\tau = C + \mu \left( \sigma_{\rm n} - P_{\rm p} \right) \tag{10}$$

where C is cohesion,  $\mu$  represents the friction coefficient,  $P_p$  is the pore pressure on the fault plane,  $\tau$  and  $\sigma_n$  are shear and normal stress acting on the fault plane, respectively. If full stress, cohesion, and friction coefficient are known, we can compute the minimal activation pressure for each focal mechanism after estimating  $\tau$  and  $\sigma_n$  from the full stress tensor. In this study, we assume no cohesion (C = 0), corresponding to a slipping fault, and an average coefficient of friction  $\mu = 0.6$  (Zoback, 1989).

Assuming the coefficient of friction is indeed 0.6, the pressure represents the minimum activation pressure, corresponds to the minimum pressure required to activate slip on pre-existing faults under the current stress conditions. Therefore, the inverted pressure represents a lower estimate of pore pressure at the time of and location of the seismic event.

### 4 Stress Field Analysis



In the strike-slip faulting regime, the stress tensor has the maximum principal stress ( $S_1$ ) in the horizontal direction ( $S_{Hmax}$ ), the intermediate principal stress ( $S_2$ ) is vertical ( $S_v$ ), and the minimal principal stress ( $S_3$ ) is also horizontal ( $S_{hmin}$ ) perpendicular to the maximum principal stress. The Decatur basin is likely in the strike-slip faulting regime based on historical seismicity data collected by the U.S. Geological Survey (USGS) and source inversions from larger events in the area.  $S_v$  and ISIP are used as additional constraints in the stress inversion as their magnitudes are easiest to measure and available from previous studies representative for the basin.

#### 4.1 Full stress inversion

- Initially, we assume the stress field is homogenous at the depth range spanned by the microseismic events. For Dataset 1, an average depth of 1.95 km was used, with a vertical stress gradient ( $\nabla \sigma_v$ ) of 23 MPa/km, resulting in a vertical stress ( $S_v$ ) of 44.9 MPa. The uncertainty in this estimate is relatively low, as the minimal density variation, with the vertical stress uncertainty ( $\delta \sigma_v$ ) is estimated at 10%. The ISIP value at this depth is estimated to be 31.6 MPa (Bauer et al., 2016). The uncertainties of ISIP ( $\delta$ ISIP) also assume 10% of the value, although we know that the ISIP is less well-constrained at this depth.
- We assign uncertainty estimates to the source mechanisms ( $\delta\alpha$ ): 5° and 10° for Dataset 1, which is approximately the difference in strike angles in this dataset, and for Dataset 2, it is estimated between 10° and 15° as the seismic data from surface receivers were not well-fitted. These uncertainties align with the Kagan angles between source mechanisms of the same events.
  - To optimize inversion results, we investigate directional constraints by testing constraint angles between 75° and 90° for plunge on Dataset 1. The analysis revealed minimal impact of angle variation on activation pressure values, leading to the selection of 90°. We applied a vertical stress plunge constraint of 90° to Dataset 1 due to its similar source mechanisms, as the similarity in source mechanisms alone did not provide well-constrained stress orientation.
  - For Dataset 2, we conduct two separate inversions, one without any directional constraint and another with the same vertical stress constrained as in Dataset 1, to compare the inversion results change under different constraints.
- Table 2Table 2 and Figure 5 summarize the result of the inversion, and Dataset 1 with the vertical stress constrained provided reasonably oriented maximum horizontal stress direction consistent with previous studies. However, Dataset 2 did not result in a stress direction consistent with the strike-slip regime without the directional constraint, while when the constraint is applied,

the stress orientation is consistent with Dataset 1. The stress magnitudes resulting from Dataset 2 are significantly higher in both cases, with wider value ranges.

Considering that 23.0 MPa was the average value of injection pressure of CO<sub>2</sub> (Williams-Stroud et al., 2020; Dichiarante et al., 2021), a minimum activation significantly exceeding the value of 23 MPa is very unlikely to result from the injection and hence the minimum activation pressures required by the inversion of Dataset 1 are too high to be caused by the CO<sub>2</sub> injection. Similarly, most events in Dataset 2 would require higher minimum activation pressure, also resulting in an unlikely scenario. Therefore, we reassessed various assumptions in our stress and minimum activation pressure to figure out whether more realistic activation pressure values could be achieved.

Table 2: Input parameters for Full Stress Inversion, homogenous vertical stress gradient and resulting stress magnitudes.

|              |                           |                          | Input val                       | ues for str             | Inverted values          |           |               |                |                         |                         |                      |                                                              |
|--------------|---------------------------|--------------------------|---------------------------------|-------------------------|--------------------------|-----------|---------------|----------------|-------------------------|-------------------------|----------------------|--------------------------------------------------------------|
| Data<br>-set | Directional constrain [°] | Average<br>Depth<br>[km] | ∇σ <sub>v</sub><br>[MPa/<br>km] | σ <sub>v</sub><br>[MPa] | δσ <sub>v</sub><br>[MPa] | δα<br>[°] | ISIP<br>[MPa] | δISIP<br>[MPa] | S <sub>I</sub><br>[MPa] | S <sub>2</sub><br>[MPa] | S <sub>3</sub> [MPa] | Minimal Activation<br>Pressure Range for<br>all events [MPa] |
| 1            | 90                        | 1.95                     | 23                              | 44.9                    | 4.49                     | 5         | 31.6          | 3.16           | 49.4                    | 44.9                    | 31.6                 | 23.3 - 26.0                                                  |
| 1            | 90                        | 1.95                     | 23                              | 44.9                    | 4.49                     | 10        | 31.6          | 3.16           | 49.1                    | 44.9                    | 31.6                 | 23.3-26.0                                                    |
| 2            | -                         | 2.54                     | 27                              | 68.8                    | 6.88                     | 10        | 31.6          | 3.16           | 81.8                    | 51.2                    | 31.6                 | 10.9 - 45.1                                                  |
| 2            | -                         | 2.54                     | 27                              | 68.8                    | 6.88                     | 15        | 31.6          | 3.16           | 81.8                    | 51.2                    | 31.6                 | 11.0 - 45.1                                                  |
| 2            | 90                        | 2.54                     | 27                              | 68.8                    | 6.88                     | 10        | 31.6          | 3.16           | 88.0                    | 68.8                    | 31.6                 | 5.3-84.0                                                     |
| 2            | 90                        | 2.54                     | 27                              | 68.8                    | 6.88                     | 15        | 31.6          | 3.16           | 88.0                    | 68.8                    | 31.6                 | 5.3-84.0                                                     |

Figure 5: Principal stresses orientations of (a) Dataset 1, with 5° uncertainty, vertical stress constraint applied (b) Dataset 2, with 10° uncertainty, no directional constraint (c) Dataset 2, with 10° uncertainty, vertical stress constraint applied. The red, green, and blue triangles represent S<sub>1</sub>, S<sub>2</sub> and S<sub>3</sub> respectively.

## 4.2 Minimal activation pressure estimation




In Dataset 1, there seems to be no significant trend in activation pressure; only events 3, 4 and 7 have relatively low activation pressure. Event 16 was excluded from the analysis as it does not belong to the strike-slip regime (Fig. 6).

Figure 6: Estimation of the minimal activation pressure of each event in Dataset1.



Analysis of Dataset 2 reveals significant spatial variation in the minimal activation pressures which seems to be nearly random (Fig. 7). The largest cluster of seismic events, located in the upper portion of the map, corresponds to regions with much higher activation pressures than those estimated for Dataset 1.

Figure 7: Estimation of the minimal activation pressure of each event in Dataset 2.

## 4.3 Refinement of Dataset2 by excluding inconsistent focal mechanisms

Several events were found to deviate from the expected strike-slip stress regime. To maintain consistency, events 6, 8, 18 and 23 were excluded from the analysis as they display focal mechanisms indicative of normal or reverse faulting, suggesting they are not governed by the same stress conditions. After filtering out these non-strike-slip events, we recalculated the minimal activation pressure for the remaining events. The values show a refined set of pressures that better align with the strike-slip faulting model. The updated stress orientation is shown in Figure 8(a).

Table 3 Minimal activation pressures calculated for the refined events in Dataset 2.

|                  |                            | ]                        | Inverted values                 |                         |                                                       |           |               |                |                         |                         |                         |                                                              |
|------------------|----------------------------|--------------------------|---------------------------------|-------------------------|-------------------------------------------------------|-----------|---------------|----------------|-------------------------|-------------------------|-------------------------|--------------------------------------------------------------|
| Number of events | Directional constraint [°] | Average<br>Depth<br>[km] | ∇σ <sub>v</sub><br>[MPa/<br>km] | σ <sub>v</sub><br>[MPa] | $\delta\sigma_{\scriptscriptstyle  m V} \ [{ m MPa}]$ | δα<br>[°] | ISIP<br>[MPa] | δISIP<br>[MPa] | S <sub>I</sub><br>[MPa] | S <sub>2</sub><br>[MPa] | S <sub>3</sub><br>[MPa] | Minimal Activation<br>Pressure Range for<br>all events [MPa] |

| 22 | 90 | 2.54 | 27 | 68.8 | 6.88 | 10 | 31.6 | 3.16 | 72.2 | 68.8 | 31.6 | 12.8 – 71.6 |
|----|----|------|----|------|------|----|------|------|------|------|------|-------------|
| 22 | 90 | 2.54 | 27 | 68.8 | 6.88 | 15 | 31.6 | 3.16 | 73.2 | 68.8 | 31.6 | 12.8 - 71.6 |

Figure 8(b) displays the spatial distribution of seismic events from the refined Dataset 2, with activation pressures indicated by a color scale ranging from 15 to 70 MPa. The main cluster of events is concentrated in the northwestern part, predominantly showing moderate to high activation pressures. Several isolated events appear in the southeastern region, including events 1, 2, 4 and 26 with the lowest activation pressures, and events 3 and 5 with the highest activation pressures. Overall, the activation pressure range of the refined dataset is lower than that of the initial dataset. Filtering events based on stress orientation appears to improve inversion accuracy.

Figure 8: (a) Stress orientation of refined Dataset 2, with vertical stress ( $S_2$ , green triangle) constraint. The red and blue triangles represent  $S_1$  and  $S_3$  respectively. (b) Estimation of the minimal activation pressures of Dataset 2 excluding the inconsistent events, with vertical stress constraint.

#### 4.4 Evaluation of various assumptions in the inversion



We evaluate various assumptions in the inversion process to explore ways to lower the minimum activation pressure values to levels similar to the average injection pressure. Reasonable adjustments to the vertical stress gradient have minimal impact on the minimum activation pressures. Similarly, uncertainties in the source mechanism orientations do little to reduce the minimum activation pressures, as they primarily affect stress orientation rather than magnitude.

However, we observed higher sensitivity of the minimum activation pressure to the estimated ISIP values. For Dataset 1, when ISIP was reduced below 29 MPa, the activation pressures remained within a reasonable range, with maximum values staying below the injection pressure. For Dataset 2, however, reducing ISIP led to unstable results, with activation pressures in some cases reaching zero or even negative values.

Table 4 Summary of optimized ISIP, stress components, and activation pressure ranges for Datasets 1 and 2.

| Dataset | Optimized<br>ISIP [MPa] | δISIP<br>[MPa] | σ <sub>v</sub> [MPa] | δα [°] | S <sub>1</sub> [MPa] | $S_2[MPa]$ | S <sub>3</sub> [MPa] | Minimal Activation Pressure Range of All Events [MPa] |
|---------|-------------------------|----------------|----------------------|--------|----------------------|------------|----------------------|-------------------------------------------------------|
| 1       | 30                      | 3.0            | 44.9                 | 5      | 49.7                 | 44.9       | 30.0                 | 20.7 -23.3                                            |
|         | 29                      | 2.9            |                      |        | 50.0                 | 44.9       | 22.9                 | 19.2 - 21.9                                           |
|         | 28                      | 2.8            |                      |        | 50.4                 | 44.9       | 28.0                 | 17.4 - 20.2                                           |
|         | 25                      | 2.5            |                      |        | 51.3                 | 44.9       | 25.0                 | 12.5 - 16.1                                           |
| 2       | 30                      | 3.0            | 68.8                 | 10     | 88.9                 | 68.8       | 30.0                 | 2.9 - 84.1                                            |
|         | 25                      | 2.5            |                      |        | 91.5                 | 68.8       | 15                   | -5.9 – 86.7                                           |

## 4.5 Evaluation of frictional coefficient for activation pressure

From Equation (10), apart from the normal and shear stresses acting on the fault, the activation pressure is also controlled by the frictional coefficient of the rock. Thus, we test the sensitivity of the inversion results by varying the frictional coefficient, as Table 5 shows. The coefficient is set at values of 0.4 and 0.6 for both Dataset 1 and Dataset 2. Lower frictional coefficient yields significantly decreased minimal activation pressure values in both datasets.

Table 5 Summary of varying frictional coefficient, and activation pressure ranges for Datasets 1 and 2.

|              |                            | Inp                    | Inverted values                 |                         |                          |           |               |                |                         |                         |                         |                                                        |
|--------------|----------------------------|------------------------|---------------------------------|-------------------------|--------------------------|-----------|---------------|----------------|-------------------------|-------------------------|-------------------------|--------------------------------------------------------|
| Data<br>-set | Directional constraint [°] | Frictional coefficient | Vσ <sub>v</sub><br>[MPa/<br>km] | σ <sub>v</sub><br>[MPa] | δσ <sub>v</sub><br>[MPa] | δα<br>[°] | ISIP<br>[MPa] | δISIP<br>[MPa] | S <sub>I</sub><br>[MPa] | S <sub>2</sub><br>[MPa] | S <sub>3</sub><br>[MPa] | Minimal Activation Pressure Range for All Events [MPa] |
| 1            | 90                         | 0.4                    | 27                              | 68.8                    | 6.88                     | 10        | 31.6          | 3.16           | 72.2                    | 68.8                    | 31.6                    | 16.7 – 21.9                                            |
|              |                            | 0.6                    |                                 |                         |                          |           |               |                |                         |                         |                         | 23.3 - 26.0                                            |
| 2            | 90                         | 0.4                    | 27                              | 68.8                    | 6.88                     | 15        | 31.6          | 3.16           | 88.0                    | 68.8                    | 31.6                    | -16.0 - 82.0                                           |
|              |                            | 0.6                    |                                 |                         |                          |           |               |                |                         |                         |                         | 5.3-84.0                                               |

#### 5 Discussion



The stress field and minimal activation pressure estimation in the study area are derived using the source mechanism data from microseismic activities and measured stress values. By applying Full Stress Inversion (FSI), we inverted the stress tensor using additional constraints on the ISIP and average vertical stress ( $S_v$ ). The minimal activation pressure of each event was then estimated by the Mohr-Coulomb fault failure criterion, assuming zero cohesion and a frictional coefficient of 0.6.

Initially, both  $S_v$  and ISIP were calculated using the stress gradient summarized from previous papers and events' average depth, which results in similar values between  $S_I$  and  $S_3$ , as well as extremely high minimal activation pressures. To address this issue,

we have applied fracture stress measurement values from published literature (Bauer et al., 2016), where the ISIP was measured at approximately 31.6 MPa, while maintaining the same vertical stress due to its lower uncertainty compared to *Shmin*. A 10% uncertainty was estimated for both values to reflect the variations in published results.

To obtain a reasonable orientation of the stress field, we assume a vertical intermediate stress orientation for both datasets to maintain consistency with the strike-slip regime observed at the Decatur site. The need for this constraint probably results from the small differences in activated faults orientations in Dataset 1 and potential inaccuracies in source mechanisms in Dataset 2. With this constraint, the inverted orientation of the maximum principal stress in both datasets aligns with published studies, showing a strike of N70°E.

As the average injection pressure was around 23 MPa, the minimal activation pressure for each event should be similar to this value for fault slip to occur. The initial results show that, generally, Dataset 1 requires activation pressure more similar to 23 MPa, especially when compared to Dataset 2. Even considering a higher uncertainty of source mechanisms do not help to reduce the minimum activation pressure in Dataset 2. When comparing the same events in Datasets 1 and 2, the source mechanisms exhibited very different fault orientations. Therefore, we tried to remove source mechanisms inconsistent with the homogenous stress field from Dataset 2. The refined results show a slightly lower range of minimum activation pressures, though some events still require unrealistically high minimal activation pressures. Thus, we may conclude that Dataset 2 does not help us to understand the stress field, most likely due to large errors in the inverted source mechanisms. A few inverted source mechanisms in Dataset 2 are not consistent with homogeneous stress field implying either stress heterogeneity or erroneous source mechanism.

The simple analysis of the minimum activation pressure leads to the requirement of a very high minimum activation pressure that far exceeds the injection pressure. Yet, we know the microseismicity occurred and was activated by the injection. Therefore, we investigate which parameter of our minimum activation pressure inversion can influence the values most significantly and result in acceptable activation pressure. We have carried out a thorough investigation and concluded that three assumed input parameters can significantly affect the inversion: ISIP (or minimum magnitude of stress), maximum horizontal stress magnitude and coefficient of friction. In the following we will discuss their effects in detail.

#### 5.1 Stress input parameter – ISIP or minimum stress magnitude estimate






Table 4 shows that reducing ISIP values resulted in an increase in the maximum principal stress ( $S_I$ ) and a decrease in the minimal activation pressure. Note that activation pressure is negative for the input parameters in the last line of Table 4. Such values are obviously not real and it means that some faults would be unstable with no pore pressure, or that natural seismic events would occur on these faults without any injection. Actually, all results with very low pore pressure values would suggest naturally occurring seismicity. But this is not observed in field and it is not correct. If the ISIP input estimate was decreased to values below 29 MPa, which is only 2.1 MPa lower than the initial estimate and well within the uncertainty of the measurement,

the calculated minimum activation pressures for all events fall below the 23 MPa for Dataset 1, which is consistent with the injection. The minimum activation pressure for events from Dataset 2 also decreases dramatically but remains very high for some events of Dataset 2. But we can explain this observation that this apparent discrepancy can result from errors in the inverted source mechanisms. Thus, we can safely conclude that lower ISIP values in the depth of the observed microseismic events can explain the triggered seismicity.

Previous studies (Bauer, 2018; Bauer et al., 2016; Lahann et al., 2017) estimated *S<sub>hmin</sub>* using formation strength tests results such as formation integrity tests and leak-off tests. However, when ISIP is estimated using these pressure values, there is a potential for overestimation ISIP estimation based on FIT/LOT-derived pressures may systematically overestimate the actual fracture closure stress, as these tests typically measure the peak pressure needed to initiate fractures (White et al., 2002; Zoback, 2007). Additionally, since these tests were conducted either at shallower depth or in various locations with different formations (Bauer, 2018), they may not provide an accurate estimate of *S<sub>hmin</sub>* for the area and depth of our datasets. Given the possibility of ISIP overestimation and the limited rock strength data availability at the target depth and formation, high uncertainties exist in stress estimation, suggesting that a lower ISIP in the reservoir is a probable scenario in the Decatur area.

# 5.2 Stress – $S_{Hmax}$





Published studies, such as Lahann et al. (2017) and Langet et al. (2020), highlight that the  $S_{Hmax}$  has the highest uncertainty, primarily due to the challenge in directly measurement it and the limited availability of indirect measurements for other parameters used to estimate the  $S_{Hmax}$ . The stress gradient for  $S_{Hmax}$  can vary from 40 MPa/km to over 80 MPa/km (Lahann et al., 2017), suggesting that at a depth of around 2 km, the stress magnitude could range from 80 MPa to over 160 MPa. This wide range leads to considerable uncertainty in the results. Using the source mechanisms of observed microseismic events and the above-discussed full stress inversion, the maximum stress magnitude in Dataset 1 was approximately 50 MPa, which is significantly below the estimated value from the stress gradient. Dataset 2, with a higher average depth of located seismic events, the inverted maximum nearly horizontal stress was 88 MPa, still at the lower bound of the range estimated by Lahann et al. (2017). We investigated fault stability values for events of Dataset 1 and found out that with the ISIP values of 31 MPa, we need maximum horizontal stress values exceeding 120 MPa to trigger seismicity with 23 MPa pore pressure at these faults. These values of maximum horizontal pressure are within the uncertainty of the estimates from previous studies, but they do not seem to be consistent with the stress inversion. We consider this is a possible, but a little bit less likely scenario.

## **5.3 Fault property**

We assumed the frictional coefficient of 0.6 during the calculations. From Equation (10), it is evident that with constant normal and shear stresses, a lower frictional coefficient results in lower minimal activation pressures. When the frictional coefficient was decreased to 0.4, we observed a significant reduction in the calculated minimum activation pressure range consistent with the injection pressures.

Laboratory and field studies (Marone, 1998) show that frictional properties can evolve during fluid injection due to changes in pore pressure and other interactions within the fault zone. Such effects, while not directly constrained in our study, may influence fault slip behavior. Our study focused on seismic events far from the injection where pressure or velocity changes due to the injection are very limited, making significant evolution of frictional properties unlikely in this case. Additionally, we set the cohesion of each fault to zero by default. We note that a non-zero cohesion would only increase the minimum activation pressure. To achieve activation pressures consistent with the injection of 23 MPa, it is likely that the cohesion of the activated faults is close to zero. We conclude that a lower coefficient of friction can also explain the observed microseismicity in Decatur.

## 5.4 Comparison with Previous Inversion Approaches

Previous methodologies for full stress inversion methods using microseismic data either excluded pore pressure effect (e.g., Reches, 1987) or relied on additional conventional data (e.g., Yin and Cornet, 1994). We developed the full stress inversion method that integrates all uncertainties of input data, such as uncertainties in source mechanisms and magnitude of stress, and invert full stress while accounting for the uncertainties simultaneously. This methodology should provide a more robust and reliable solution.

#### 425 6 Conclusions



415

The IBDP site is a well-known CCS project with extensive seismic monitoring data, located in a region predominantly characterized by a strike-slip regime. In this setting, the maximum horizontal stress acts as the maximum principal stress, and the intermediate principal stress is vertical. In this study, we applied the full stress inversion approach to two different source mechanism datasets, combined with ISIP and vertical stress magnitude as the additional magnitude constraints, to invert the principal stress orientation and magnitudes in the target area. Using the inverted stress, we applied the Mohr-Coulomb failure criterion to estimate the minimal activation pressure of each event across both datasets.

Initial minimum activation pressures estimated with the observed data yielded unrealistically high activation pressures, significantly higher than the average injection pressure. The inverted principal stress orientation, with an  $S_{Hmax}$  strike of N70°E, aligns with previous studies, but the intermediate stress orientation had to be constrained to be vertical. Higher quality source mechanisms (Dataset 1) resulted in more consistent stress orientation.

Further analysis shows that reduced ISIP or lower coefficient of friction significantly lowered minimal activation pressures resulting in activation pressures consistent with the injection pressures. Alternatively, consistent activation pressures can be achieved with large  $S_{Hmax}$  values, however, our stress inversion resolves  $S_{Hmax}$  within lower ranges in both datasets, inconsistent with the need for extremely high  $S_{Hmax}$  values required to trigger seismicity. This implies that the high  $S_{Hmax}$  value estimation,

though theoretically possible, lacks empirical support from measurements. Consequently, the reduced ISIP solution provides a more reasonable explanation consistent with all available constraints.

Our findings demonstrate that well-constrained source mechanisms provide reliable stress field information, highlighting the importance of high-quality monitoring in geological storage projects. Stress field characteristics and fault properties appear to significantly influence activation pressure estimation, when negligible cohesion is assumed for the faults. This study emphasizes the critical need for accurate ISIP measurements that can lead to more reliable  $S_{Hmax}$  estimates in the reservoir to ensure reduced uncertainty for estimates of stress state from moment tensor inversions in areas where well data are sparse or not available, for improved prediction of the long-term reservoir response to injection.

## Data availability

445

The datasets analyzed in this study are publicly available as part of previously published works. Dataset 1 is available in the supplementary material of Langet et al. (2020), containing timing, magnitude, location including uncertainties of selected microseismic events. Dataset 2 is sourced from the electronic supplement of Williams-Stroud et al., (2020), the dataset was previously analyzed internally by Staněk et al. (2019, unpublished).

#### **Author contributions**

LE, ZJ, VV and TG conceptualized the study and designed the research approach. DA, LE, ZJ and UBW developed the stress inversion methodology. TG, along with LE and ZJ, performed the calculations and data analysis. The initial manuscript draft was primarily written by TG. LE, ZJ, SCW, and VV provided feedback and contributed to manuscript revisions. All authors reviewed and approved the final version.

## **Competing interests**

The contact author has declared that none of the authors has any competing interests.

## 460 Acknowledgements

The authors acknowledge funding from the EU Horizon Europe Research and Innovation Program through the Doctoral Network of the Marie Skłodowska-Curie Actions SMILE (https://smile-msca-dn.eu/), under grant agreement 101073281. IMEDEA is an accredited "Maria de Maeztu Excellence Unit" (Grant CEX2021-001198, funded by MICIU/AEI/10.13039/501100011033).

#### 465 Financial support

This research has been funded by the EU Horizon Europe Research and Innovation Program through the Doctoral Network of the Marie Skłodowska-Curie Actions SMILE (https://smile-msca-dn.eu/), under grant agreement 101073281. IMEDEA is an accredited "Maria de Maeztu Excellence Unit" (Grant CEX2021-001198, funded by MICIU/AEI/10.13039/501100011033).

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
