# Peer review of "Constraints on Stress Tensor from Microseismicity at Decatur"

_EGUsphere, 2025_

## Author Comment (AC1)

***Reply to the RC1's comment***

*'The article presents a study on constraints on stress tensor on the monitored seismicity at the Illinois Basin Decatur Project (IBDP), a $CO_2$ sequestration project effective between 2011 and 2014. The study was encouraged due to the large uncertainty on the stress tensor magnitudes in the Decatur reservoir. Its aim is to refine the stress field estimations by the full stress inversion method from focal mechanisms of microseismic events from two datasets and by considering borehole data: the values of the instantaneous shut-in pressure and the average of vertical stress. From the update of stress tensor, they compute an updated value of the minimum activation pressure at the project site, significantly smaller than the originally estimated threshold pressures for the regional faulting network. The manuscript addresses the importance of the estimation of the minimal activation pressure, one of the main parameters in the design of the injection protocol, and presents a methodology that refines the estimation of the stress tensor.*

*Overall, the research topic and the proposed methodology are significant. The manuscript is clear, and presents a literature review of the stress estimations of the study case. The methods and results are well explained. Yet, I would encourage the authors to revise the writing of the manuscript. Indeed, the manuscript could benefit from a little upgrade in its writing (there is a lot of repetitions of certain words along the sentences, and the transitions between paragraphs and sections can be quite abrupt).'*

Thank you for the positive feedback, we acknowledge the comment regarding the writing style, particularly the word repetitions and transitions. We have revised the manuscript accordingly to improve the clarity and rephrasing repetitive parts for a better readability.

*'Minor comments:*

*in 1. Introduction, there is only one subheading. The subheading might be deleted. In Introduction: The introduction presents stress estimations from previous studies (section 1.1.), but an introduction of the study case (currently in 2.1.) would be expected to be done before.'*

We have moved the description of the IBDP site (previous 2.1) to the Introduction as Section 1.1, and accordingly, the previous Section 1.1 (stress state) is now Section 1.2. Section 2 has been revised to contain only the dataset description, and the previous Section 2.2 has been updated accordingly. With this we believe it is better to keep the subheading of 1.1 on line 49 to differentiate it from 1.2 on line 79. We hope these changes enable the reader to orient in the text.

*'Lines 90-93: "This is an important constrain on our pore pressure limitation – the pressure that activated induced seismicity cannot significantly exceed this value because we know the injection activated this seismicity and this is the pressure that caused the seismicity to occur*

*(we can neglect possible thermal effects; selected induced events are far from the injection well)."
The sentence is quite long and not very fluid.'*

Thank you, we agree the sentence was not clear, we have rewritten it for clarity:

"The average injection pressure provides an important constraint on the pore pressure, as the pore pressure that activated induced seismicity is not expected to significantly exceed this value. We know that the injection triggered these events and the events are far from the injection well. Therefore, thermal effects can be neglected as triggering mechanism. Hence, the most likely triggering mechanism is increased pore pressure on pre-existing faults/fractures."

*'Line 115: "We use the fact that this seismicity exists (i.e. the stress state or pore pressure in the reservoir was perturbed to induced seismicity)". What do the authors mean by this?'*

We have revised this sentence to clarify it:

"We interpret the occurrence of induced seismicity as evidence that the stress state or pore pressure in the reservoir was sufficiently perturbed to trigger slip. Additionally, we use the source mechanisms of these events to further constrain the stress state in the Decatur Basin."

*'Line 119: "small number", how many?'*

Thank you for the comment, we have clarified the text to specify the number of events: 39 unique microseismic events in total (23 from dataset1 and 26 from dataset2 with 11 events overlapping between the two datasets).

*'Lines 127-… : the bullet points are a bit rough, it could be better to reformulate as paragraphs'*

We have restructured the bullet points into two paragraphs to improve the paper style.

*'Line 140: "parameters of some seismic events differ from a dataset to the other", how much? Is it questioning the validity of the seismic interpretation of one dataset?'*

We are sorry but we cannot find the text ': "parameters of some seismic events differ from a dataset to the other". We quantified the discussion of differences between lines 140 and 145. Partly it may question validity of inverted mechanisms as it is discussed later. We prefer to discuss the validity of the dataset at the Discussion section.

*'Lines 150-…: "To determine the pore pressure that caused induced seismicity, we need to know the shear and normal stress on the fault plane, the coefficient of friction and the cohesion on the faults of microseismic events. To determine shear and normal stress on a fault, we need to know the full stress tensor." The two sentences have the same structures and read like a repetition.'*

We have rephrased the sentences to avoid repetition and improve readability as:

"To determine the pore pressure responsible for inducing seismicity, it is necessary to know the shear and normal stresses on the fault plane, along with the friction coefficient and cohesion of the faults associated with microseismic events. Calculating the shear and normal stresses on a fault plane requires knowledge of the full stress tensor."

*'Line 237: typo in "compute"'*

Corrected, thank you.

*'Line 240 "the pressure represents the minimum activation pressure, which is the minimum pressure needed to activate the fault". the sentence feels repetitive.'*

We have rewritten this sentence to:

"… which corresponds to the minimum pressure required to activate slip on pre-existing faults under the current stress conditions."

*'In generally, the colorbars should be wider in the figures, the colors are difficulty distinguishable.'*

Thanks for pointing this out, we have increased the width of the colorbars in all relevant figures to enhance visibility.

*'In Table 4, one Minimal Activation Pressure Range of All Events is negative. What does it mean for the methodology?'*

Thank you for catching this, the negative value means that the fault is unstable in the selected stress field even with 0 MPa pore pressure. This would mean the seismicity would be naturally occurring in the area independently of the injection. This is obviously not correct, hence either ISIP or dISIP is not correct. We explain in the text.

*'Line 354: what is "maximum horizontal press magnitude"?'*

Thanks for the correction, it was a typo, we have corrected it to 'maximum horizontal stress magnitude'.

*'In the manuscript, I would suggest to use the present tense when describing the analyses made by the authors in this study.'*

Thank you for pointing this out, we have changed the time tense of the analysis part in this study.

---

## Author Response (AR1)

**Response to Reviewers - Manuscript egusphere-2025-1384**

**Reviewer 1**

**Comment 1.1:**

The article presents a study on constraints on stress tensor on the monitored seismicity at the Illinois Basin Decatur Project (IBDP), a CO2 sequestration project effective between 2011 and 2014. The study was encouraged due to the large uncertainty on the stress tensor magnitudes in the Decatur reservoir. Its aim is to refine the stress field estimations by the full stress inversion method from focal mechanisms of microseismic events from two datasets and by considering borehole data: the values of the instantaneous shut-in pressure and the average of vertical stress. From the update of stress tensor, they compute an updated value of the minimum activation pressure at the project site, significantly smaller than the originally estimated threshold pressures for the regional faulting network. The manuscript addresses the importance of the estimation of the minimal activation pressure, one of the main parameters in the design of the injection protocol, and presents a methodology that refines the estimation of the stress tensor.

Overall, the research topic and the proposed methodology are significant. The manuscript is clear, and presents a literature review of the stress estimations of the study case. The methods and results are well explained. Yet, I would encourage the authors to revise the writing of the manuscript. Indeed, the manuscript could benefit from a little upgrade in its writing (there is a lot of repetitions of certain words along the sentences, and the transitions between paragraphs and sections can be quite abrupt).'

**Response:**

Thank you for the positive feedback, we acknowledge the comment regarding the writing style, particularly the word repetitions and transitions. We have revised the manuscript accordingly to improve the clarity and rephrasing repetitive parts for a better readability.

**Comment 1.2:**

in 1. Introduction, there is only one subheading. The subheading might be deleted. In Introduction: The introduction presents stress estimations from previous studies (section 1.1.), but an introduction of the study case (currently in 2.1.) would be expected to be done before.'

**Response:**

We have moved the description of the IBDP site (previously Section 2.1) to the Introduction as Section 1.1, and accordingly, the previous Section 1.1 (Stress State) is now Section 1.2. Section 2 has also been revised to contain only the dataset description, with the previous Section 2.2 updated accordingly. We have streamlined the Introduction to focus primarily on

the broader context of CCS and induced seismicity, along with our study's motivation and objectives. Concurrently, all comprehensive site background, including geological descriptions, injection parameters, and monitoring infrastructure, is now cohesively and exclusively presented within the expanded Section 1.1 (Study Site). We believe these changes significantly improve the manuscript's organization and enable the reader to better navigate the text.

**Comment 1.3:**

'Lines 90-93: "This is an important constrain on our pore pressure limitation – the pressure that activated induced seismicity cannot significantly exceed this value because we know the injection activated this seismicity and this is the pressure that caused the seismicity to occur (we can neglect possible thermal effects; selected induced events are far from the injection well)." The sentence is quite long and not very fluid.'

**Response:**

Thank you, we agree the sentence was not clear, we have rewritten it for clarity:

"The average injection pressure provides an important constraint on the pore pressure, as the pore pressure that activated induced seismicity is not expected to significantly exceed this value. We know that the injection triggered these events and the events are far from the injection well. Therefore, thermal effects can be neglected as triggering mechanism. Hence, the most likely triggering mechanism is increased pore pressure on pre-existing faults or fractures."

**Comment 1.4:**

'Line 115: "We use the fact that this seismicity exists (i.e. the stress state or pore pressure in the reservoir was perturbed to induced seismicity)". What do the authors mean by this?'

**Response:**

We have revised this sentence to clarify it:

"We interpret the occurrence of induced seismicity as evidence that the stress state or pore pressure in the reservoir was sufficiently perturbed to trigger slip. Additionally, we analyze the source mechanisms of the events to further constrain the stress state in the Decatur Basin."

**Comment 1.5:**

'Line 119: "small number", how many?'

**Response:**

Thank you for the comment, we have clarified the text to specify the number of events: 39 unique microseismic events in total (23 from dataset1 and 26 from dataset2 with 11 events overlapping between the two datasets).

**Comment 1.6:**

'Lines 127-...: the bullet points are a bit rough, it could be better to reformulate as paragraphs'

**Response:**

We have restructured the bullet points into two paragraphs to improve the paper style.

**Comment 1.7:**

'Line 140: "parameters of some seismic events differ from a dataset to the other", how much? Is it questioning the validity of the seismic interpretation of one dataset?'

**Response:**

We are sorry but we cannot find the text ': "parameters of some seismic events differ from a dataset to the other". We quantified the discussion of differences between lines 140 and 145. Partly it may question validity of inverted mechanisms as it is discussed later. We prefer to discuss the validity of the dataset at the Discussion section.

**Comment 1.8:**

'Lines 150-...: "To determine the pore pressure that caused induced seismicity, we need to know the shear and normal stress on the fault plane, the coefficient of friction and the cohesion on the faults of microseismic events. To determine shear and normal stress on a fault, we need to know the full stress tensor." The two sentences have the same structures and read like a repetition.'

**Response:**

We have rephrased the sentences to avoid repetition and improve readability as:

"To determine the pore pressure responsible for inducing seismicity, it is necessary to know the shear and normal stresses on the fault plane, along with the friction coefficient and cohesion of the faults associated with microseismic events. Calculating the shear and normal stresses on a fault plane requires knowledge of the full stress tensor."

**Comment 1.9:**

'Line 237: typo in "compute"

**Response:**

Corrected, thank you.

**Comment 1.10:**

'Line 240 "the pressure represents the minimum activation pressure, which is the minimum pressure needed to activate the fault". the sentence feels repetitive.'

**Response:**

We have rewritten this sentence to:

"... which corresponds to the minimum pressure required to activate slip on pre-existing faults under the current stress conditions."

**Comment 1.11:**

'In generally, the colorbars should be wider in the figures, the colors are difficulty distinguishable.'

**Response:**

Thanks for pointing this out, we have increased the width of the colorbars in all relevant figures to enhance visibility.

**Comment 1.12:**

'In Table 4, one Minimal Activation Pressure Range of All Events is negative. What does it mean for the methodology?'

**Response:**

Thank you for catching this, the negative value means that the fault is unstable in the selected stress field even with 0 MPa pore pressure. This would mean the seismicity would be naturally occurring in the area independently of the injection. This is obviously not correct, hence either ISIP or dISIP is not correct. We explain in the text.

**Comment 1.13:**

'Line 354: what is "maximum horizontal press magnitude"?'

**Response:**

Thanks for the correction, it was a typo, we have corrected it to 'maximum horizontal stress magnitude'.

**Comment 1.14:**

'In the manuscript, I would suggest to use the present tense when describing the analyses made by the authors in this study.'

**Response:**

Thank you for pointing this out, we have changed the time tense of the analysis part in this study.

**Reviewer 2:**

**Comment 2.1**

This manuscript presents is a study about stress state estimations, which are crucial for determination of the minimal stress state of induced seismicity related to fluid injection. It is case study on the Illinois Basin Decatur Project (IBDP). Authors tackle the problem of stress calculation based on focal mechanism and additional stress parameters. Authors unveil the influence of the focal mechanism uncertainties as well as other crucial parameters such as friction coefficient and instantaneous shut-in pressure and the average of vertical stress for estimation minimum activation stress/pressure. The latter parameter is crucial in designing the injection operations and control the induced seismicity. I find this manuscript well designed and written. Authors clearly state the scientific problem and show possible solutions within assumed model and parameters.

**Response:**

We greatly appreciate the positive evaluation from the reviewer.

**Comment 2.2:**

However, I have one general comment and some minor ones.

You assumed constant friction during the whole injection and afterwards. What about friction coefficient changes due to injection? Maybe it would be informative to check the time evolution of the seismicity and injection and compare it with assumed permeability.

**Response:**

Thank you for raising this interesting point. The assumption of constant friction Is a simplification due to the lack of direct measurement of the friction coefficient. We have now added a discussion paragraph to address this issue.

**Change in manuscript:**

'Laboratory and field studies (Marone, 1998) show that frictional properties can evolve during fluid injection due to changes in pore pressure and other interactions within the fault zone. Such effects, while not directly constrained in our study, may influence fault slip behavior. Our study focused on seismic events far from the injection where pressure or velocity changes due to the injection are very limited, making significant evolution of frictional properties unlikely in this case. Additionally, we set the cohesion of each fault to zero by default. We note that a non-zero cohesion would only increase the minimum activation pressure. To achieve activation pressures consistent with the injection of 23 MPa, it is likely that the cohesion of the activated faults is close to zero. We conclude that a lower coefficient of friction can also explain the observed microseismicity in Decatur.'

Marone (1998) shows that the frictional changes are caused by velocity or contact time changes in the fault zone. We do not see a process that can change these more than 1 km from the injection point.

**Comment 2.3:**

I miss the analysis of temporal behavior of the seismic activity with relation to injection time. Both clusters were activated after start of the injection, but with significant delay, which may be due to fluid or pore-pressure migration as well as cumulative stress changes due to consecutive injection volumes.'

**Response:**

We thank the reviewer for this insightful comment; we agree that such an analysis would be highly valuable for understanding the underlying physical mechanisms at Decatur. However, we found that the uncertainties in event locations, particularly within Dataset 2 presented a big challenge to conducting a robust and conclusive space-temporal analysis. Pressure changes at Decatur were small, around 1 MPa at the injection well, and given the high permeability and thickness of Mt. Simon sandstone, the pressure changes at the location of the induced events are minor. Nonetheless, the triggering mechanism was likely related to pore pressure diffusion as the pressure front advanced away from the injection well and destabilized critically stressed faults.

**Comment 2.4:**

Minor remarks, which may help readers better understand the work and some questions and suggestions for authors:

Lines 62-67: This is a bit difficult to follow, when You compare stress gradient with Shmin value at particular depth, since reader may not know how these values are related to each other. I suggest to write it more straightforward and You should explain how horizontal stress gradient depend on depth in this area.

**Response:**

Thank you for raising this point. We have rephrased the paragraph to more clearly describe the relationship between stress and depth, and added a short explanation of how stress gradients are used to estimate stress magnitudes at a given depth in this area.

**Comment 2.5:**

Lines 95-100: Maybe map with sensor location would allow reader to better get the geometry of the observation setup.

**Response:**

Thanks for pointing this out. A new figure (Figure 1 (b)) has been added to show the spatial distribution of monitoring sensors and stations used in this study, including their depth when relevant.

**Comment 2.6:**

Lines 125-137: Focal mechanism quality may depend on many factors: two similar but different methods and different station setup and focal coverage may be one of these. I would like to get more info about the focal coverage and number of stations (picks) used in inversion.

**Response:**

To provide further clarity, we have extended the "Dataset" section in the revised manuscript to include more detailed information regarding the focal mechanism determination for both datasets. However, we are not able to provide information on focal coverage and number of stations used in the inversion as the authors of the cited sources did not reveal this information.

For Dataset 1 (Langet et al.,2020), the focal mechanisms were inverted using P-wave amplitude and polarities from manually picked P-waves on vertical components of both downhole and surface arrays. While the exact number of picks per event is not explicitly listed in the main publication, the supplementary material for their paper provides a comprehensive list of seismic stations used in the study (Table S1 in the Supplementary Material). Furthermore, Table S3 in their Supplementary Material details quality factors for each event's focal mechanism, indicating the distribution of stations on the focal sphere. We have provided summary information on these inversions in the modified text.

For Dataset 2 (Williams-Stroud et al., 2020), the focal mechanisms were inverted using both P- and S-wave amplitudes using data primarily from surface stations. A consistent number of 12 stations were used for each event, with the number of P-wave picks ranging from 6 to 16 and S-wave picks ranging from 6 to 16. Quantitative metrics such as azimuthal gap are not provided in the publicly available catalog for this dataset.

**Changes in manuscript:**

- '- Dataset 1: ... The locations of events were determined in a 1D velocity model, incorporating seismic data from the downhole geophone array and additional monitoring receivers from USGS and ISGS networks to improve the data accuracy. The combined use of surface and downhole sensors significantly enhances the station coverage over the focal sphere (Langet et al., 2020). The source mechanisms ...
- Dataset 2: The second dataset of 26 most suitable events located throughout the reservoir is obtained from electronic supplement of Williams-Stroud et al. (2020) (Fig. 3 (b)). Events were located using data from surface stations of USGS and ISGS, with a consistent

number of 12 stations utilized per event. The source mechanisms were inverted from both P- and S-wave amplitudes using a 1D velocity model and attenuation model (Staněk and Eisner, 2017). The number of P-wave picks for these inversions ranged from 6 to 16, and S-wave picks also ranged from 6 to 16.

Both datasets ... whereas Dataset 2 had different station configurations (Williams-Stroud et al., 2020), primarily relying on surface stations. Consequently, the focal coverage for the events in Dataset 2 mainly constrained the horizontal distribution of ray path, which may lead to less constrained vertical control and ambiguities in the focal mechanism solution. We cannot assume the depths of Dataset 1...'